

# Measurement of Atmospheric CO$_2$ Column Concentrations to Cloud Tops with a Pulsed Multi-wavelength Airborne Lidar

Jianping Mao[1], Anand Ramanathan[1], James B. Abshire[2], S. Randy Kawa[2], Haris Riris[2], Graham R. Allan[3], Michael Rodriguez[3], William E. Hasselbrack[3], Xiaoli Sun[2], Kenji Numata[2], Jeff Chen[2], Yonghoon Choi[4], Mei Ying Melissa Yang[4]

[1]Earth System Science Interdisciplinary Center, University of Maryland, College Park, MD 20740, USA
[2]NASA Goddard Space Flight Center, 8800 Greenbelt Road, Greenbelt, MD 20771, USA
[3]Sigma Space Inc., Lanham, MD 20706, USA
[4]NASA Langley Research Center, Hampton, VA 23681, USA

*Correspondence to*: Jianping.Mao@nasa.gov

**Abstract**. We have measured the column-averaged atmospheric CO$_2$ mixing ratio to a variety of cloud tops by using an airborne pulsed multiple wavelengths integrated-path, differential absorption (IPDA) lidar. Airborne measurements were made at altitudes up to 13 km during the 2011, 2013 and 2014 ASCENDS science campaigns flown in the west and mid-west United States and were compared to those from an in situ sensor. Analysis of the lidar backscatter profiles show the average cloud top reflectance was ~5% for the CO$_2$ measurement at 1572.335 nm except to cirrus clouds which had lower reflectance. The energies for 1-μs wide laser pulses reflected from cloud tops were sufficient to allow clear identification of CO$_2$ absorption line shape and then to allow retrievals of atmospheric column CO$_2$ from the aircraft to cloud tops more than 90% of the time. Retrievals from the CO$_2$ measurements to cloud tops had minimal bias but larger standard deviations when compared to those made to the ground, depending on cloud top roughness and reflectance. The measurements show this new capability helps resolve CO$_2$ horizontal and vertical gradients in the atmosphere. When used with nearby full column measurements to ground, the CO$_2$ measurements to cloud tops can be used to estimate the partial column CO$_2$ concentration below clouds, which should lead to better estimates of surface carbon sources and sinks. This additional capability of the range-resolved CO$_2$ IPDA lidar technique provides a new benefit for studying the carbon cycle in future airborne and space-based CO$_2$ missions.

## 1. Introduction

Precise and accurate atmospheric CO$_2$ measurements with global coverage and full seasonal sampling are crucial to advance carbon cycle sciences (Schimel et al., 2016). Passive remote sensing of column-averaged atmospheric CO$_2$ mixing ratio (XCO$_2$) from space using Earth's surface reflected sunlight, e.g., the U.S. Orbiting Carbon Observatory ( OCO-2; Crisp et al., 2004) and the Japanese Greenhouse gases Observation SATellite (GOSAT; Kuze et al., 2009), is limited to cloud-free pixels, where the photon path length can be well characterized. However those missions are unable to provide quality retrievals in the present of clouds and aerosols due to significant modification of the photon path length by scattering (e.g., Mao and Kawa, 2004; Houweling et al., 2005; Aben et al., 2007; Butz et al., 2009; Uchino et al., 2012; Yoshida et al., 2013; Guerlet et al., 2013). Passive remote sensing data from space thus are limited in spatial coverage and seasonal sampling, which may cause large uncertainty in regional and hemispheric carbon flux estimates (Chevallier et al. 2014; Reuter et al., 2014; Feng et al., 2009; 2016a, b).

Active (lidar-based) remote sensing of CO$_2$ from space will carry its own optical source and so will allow day and night measurements and full year sampling globally. Range resolved laser measurements allow precisely determining the photon path length and thus enables accurate retrievals of XCO$_2$ to the scattering surface, even in





the presence of thin clouds and aerosols. Because of these benefits the U.S. National Research Council recommended the NASA Active Sensing of $CO_2$ Emissions over Nights, Days, and Seasons (ASCENDS) mission in the 2007 report *Earth Science and Applications from Space: National Imperatives for the Next Decade and Beyond*.

5 NASA Goddard Space Flight Center (GSFC) has developed a pulsed multi-wavelength integrated-path, differential absorption (IPDA) lidar approach called the $CO_2$ Sounder to measure atmospheric $CO_2$ from space as a candidate for NASA's ASCENDS mission (Abshire et al., 2010, 2013, 2014, 2017). It uses a time-resolved receiver to record the altitude-resolved laser backscatter profiles at all measurement wavelengths, which enables accurate ranging to cloud tops. This allows retrieval of partial column $XCO_2$ to cloud tops in addition to those for the full column to the ground. The difference in absorption line shapes between the full column and the partial column to cloud tops can be 10 used to estimate partial column $XCO_2$ between the ground and cloud tops for lower-layer atmospheric $CO_2$ (Ramanathan et al., 2015).

The GSFC $CO_2$ Sounder has been flown on NASA DC-8 aircraft since 2010 over a variety of sites in the U.S., along with other ASCENDS airborne lidar candidates together with accurate in situ $CO_2$ sensors. This paper describes the retrievals and analyses of partial column $XCO_2$ measurements made to cloud tops for a variety of cloud types during 15 the 2011, 2013 and 2014 ASCENDS airborne campaigns.

## 2. Measurement Approach

The airborne $CO_2$ Sounder lidar uses a tunable narrow line-width laser to measure $CO_2$ absorption at 30 wavelengths across the vibration-rotation line of $CO_2$ centered at 1572.335 nm. The line has a Lorentz half-width $\alpha_L \approx 0.07$ cm$^{-1}$ 20 (~ 17 pm or 2.1 GHz) at standard atmospheric pressure and temperature. The laser is pulsed in a width of 1 μs at a rate of 10 kHz (or a step of 100 μs) and the laser scans across the $CO_2$ line at 30 wavelengths at a 300Hz rate. The wavelength separation of each pulse was 450 MHz or 0.015 cm$^{-1}$ evenly for 2011 and 2013 campaigns. The sampling spacing was changed for the 2014 campaign to be 250 MHz near line center and 2 GHz or 0.067 cm$^{-1}$ on line wings to allow for more on-line samples. The laser line-width is approximately 15 MHz or 0.005 cm$^{-1}$. This is 25 considerably higher than that of GOSAT (~0.2 cm$^{-1}$; Kuze et al., 2009), OCO-2 (~0.3 cm$^{-1}$; Crisp et al., 2004) and the ground-based Fourier Transform Spectrometers of the Total Carbon Column Observing Network (~0.02 cm$^{-1}$; Wunch et al., 2011). The narrow linewidth allows the measured $CO_2$ line shape to be fully resolved. The parameters of the GSFC $CO_2$ Sounder have been summarized in tables of previous publications (Abshire et al., 2010, 2013 and 2014).

30 The $CO_2$ Sounder is mounted in a fixed nadir-pointed orientation, which results in vertically directed measurements from the aircraft during normal horizontal flights. However, when the airplane tilts, the laser measurement direction points away-nadir and the measurement direction is accounted for in the data processing. The laser photons backscattered from the atmosphere and ground are collected by a 20-cm receiver telescope, after passing through a narrow (~ 1 nm) band-pass filter, and then are focused onto the lidar detector. The bandwidth of receiver is 10 MHz, 35 and has a response time of 30 ns. The range backscatter profiles are recorded for all laser wavelengths at a 10 Hz rate. The lidar measures range to better than 0.25 m to flat surfaces over a horizontal path from the laboratory (Amediek et al., 2013).

In the following sections, we briefly describe and illustrate GSFC $CO_2$ Sounder measurements, including backscattering, range, surface roughness and surface reflectance that enable retrievals of the partial column $XCO_2$ to 40 cloud tops

### 2.1 Backscatter Measurements




As an example Figure 1 shows a 30-minute duration of backscatter profiles measured over Iowa during the 2011 ASCENDS airborne science campaign. The figure shows height-resolved lidar returns from the ground, and from the top of fair weather cumulus clouds at the top of the planetary boundary layer (PBL) near 2-km as well as from the body of high-altitude cirrus clouds. Some distributed aerosols were present, particularly within the boundary layer but the signal was weak. The backscatter profiles at two discrete times are also shown.

## 2.2 Range Measurements and Surface Roughness

The laser pulse energies from each significant scattering surface can be processed at each of the 30 transmitted wavelengths to display $CO_2$ line absorption features in terms of optical depth (OD). An example is shown in Fig. 2. These samples of the absorption line shape may be used to retrieve $XCO_2$ from aircraft altitude to each significant scattering surface by fitting measured ODs to pre-calculated ODs for the same atmospheric state. The ranging capability of pulsed lidar allows accurate determination of photon path-length for $XCO_2$ retrievals. This is a major advantage of this lidar approach over passive approaches for remote sensing of greenhouse gases when the reflecting surface elevation is uncertain (e.g., tall trees) and when the atmosphere has significant scattering (Mao and Kawa, 2004; Aben et al., 2007).

In order to improve precision, the raw lidar measurements may be aggregated to a larger scale before used for $XCO_2$ retrievals. The range to the scattering surfaces may vary significantly within the aggregated scale, depending on the roughness of scattering surface and data aggregation time. In previous measurements (Abshire et al., 2013), the standard deviation of range measurements from airplane to a flat surface, e.g., Railroad Valley, NV, was about 1 m, but increased to 25 m over mountains within a 10-s data average time, which corresponds to 2 km ground track length. These changes are caused by changes in surface topography within the averaging time. During a flight in the 2014 campaign, one flight was made over the Pacific Ocean near California coastline with low winds. The lidar range measurements made at 10Hz show 0.5 m standard deviation, as shown in Fig. 3. The standard deviation increased to about 1 m after measurements are averaged over 5 seconds. Although the data aggregation before retrieval can increase signal-to-noise ratio (SNR) and improve retrieval precision for flat surfaces, over rougher surfaces like mountains there can be more variation in the photon path-length, which can limit the data averaging time before retrieval. Since surface roughness and $XCO_2$ variations are smaller over ocean than over land, data can be averaged over a longer time over oceans before retrieval.

The elevations of cloud tops can vary significantly. Lidar measurements showed the standard deviation of range measurements to marine stratus cloud tops from the 2014 flights at the California coastline was approximately 5 m for a 0.1-s averaging time and increased to 18 m for 5-s averages, as shown in Fig. 4, which is reasonably consistent with estimates from the 2011 flights over the Pacific Ocean (Abshire et al., 2013). As expected the range measurements to puffy popcorn-like cumulus cloud tops made in the 2014 campaign showed more variation. The standard deviation was 42 m for 0.1-s averages and 107 m for 5-s averages, as shown in Fig. 5. Thus, the partial column $XCO_2$ measurements made to cumulus cloud tops using 10-s averaged data are expected to be noisier than these over marine stratus clouds.

## 2.3 Cloud Reflectance

The lidar measurement of backscatter profiles also allows estimating the reflectance of the scattering surfaces. For a pulsed lidar, the reflectance of a scattering surface is given by

$$r_s = \frac{E_r}{E_{tr}} \frac{R^2}{\tau_{sys}} \qquad (1)$$





where $E_r$ is the signal backscatter pulse energy, $E_{tr}$ is the laser transmitter energy, $R$ is the range to the surface and $\tau_{sys}$ is the lidar system transmission. The lidar signal from an elevated surface such as an aerosol or a cloud layer only includes the backscattered component of the laser. For the pulsed $CO_2$ Sounder, only the photons backscattered by clouds within the 150-m thick atmospheric layer (with the 1-μs laser pulse width) are collected and then used to

estimate cloud reflectance. In contrast for cloud illuminated by sunlight, a passive sensor viewing the cloud collects all photons including those scattered from outside of field-of-view, as well as photons scattered forward by cloud particles and then backscattered by lower clouds. Thus, for thick clouds more sunlight is returned and the passive measured cloud reflectance is much higher at these wavelengths.

Figure 2 shows an example of airborne lidar measurements and the relative strength of pulse echoes reflected from

the ground, cumulus and cirrus clouds. The echoes from the ground show the sharpest vertical profile as ground is a solid surface. The vertical extent of backscatter from cirrus clouds is broader than those from cumulus cloud tops. This is because cirrus clouds were semi-transparent while cumulus clouds were denser so that only photons reflected back from the cloud tops are scattered back to the receiver. For cumulus clouds, the peak pulse return at off-line wavelengths (in red) was about 40% of the ground return while for cirrus clouds the peak return was approximately

25% of ground return.

The lidar-measured cloud top reflectance values were calculated for each flight of these campaigns. Figure 6 shows that for the cumulus clouds over Iowa in 2014, after averaging in 150-m vertical layers and over 10-s of ground track, the median value of cloud top reflectance was approximately 5%. The averaged reflectance of Pacific marine stratus cloud tops during the 2011 and 2014 flights was about 4%. The reflectance of the dense and tall

cumulonimbus clouds during a thunderstorm in a 2014 flight in Iowa was slightly higher, 6%, while the ground reflectance was estimated to be 20%. The range resolved reflectance of the cirrus clouds was found to be substantially lower, depending on physical and spatial structure of the clouds. As shown in the backscatter vertical profiles in Fig. 1, after lidar range correction, reflectance of relatively dense and thick cirrus cloud on the left bottom panel (UTC 22:40:04) was half of cumulus cloud reflectance or 2~3% while reflectance of the thinner cirrus clouds

on the right panel (UTC 23:00:24) was 1%. For the distributed backscatter from cirrus clouds, if the vertical signal accumulating layer is increased, then the integrated pulse echo energy and reflectance would be higher.

Data analysis shows that the pulsed lidar signals from cloud tops were sufficient to clearly capture the $CO_2$ absorption line shape. The full line shape from the total of 30 wavelengths across the line is shown in Fig. 2. With the lidar range measurement, this allows quality retrievals of $XCO_2$ to cloud tops. These retrievals are expected to be

noisier than those to the ground due to the lower reflectance of clouds. During the 2013 and 2014 campaigns in the west and mid-west United States, the ground reflectances were from 15-40% (listed in Table 2). Meanwhile, the reflectance of ocean surface at nadir was 10-20%, depending on wind speed, and quickly dropped to nearly zero when aircraft tilts and laser points to off-nadir. Snow ice particles have a strong absorption band near 1500 nm and snow surfaces have reflectances of 2-10% at 1572 nm wavelengths, depending on snow condition, e.g., grain size

(Wiscombe and Warren, 1980; Painter and Dozier, 2004). In the campaign, snow scenes were sometimes mixed with other more highly reflecting objects, e.g., trees and rocks. Note reflectance of 40% over desert surfaces is an established standard for estimating reflectance and is very close to what in situ measurements made by the GOSAT validation team in Railroad Valley at $CO_2$ Sounder measurement wavelengths (Kuze et al., 2011).

## 3.  Cloud Identification and Data Processing

### 3.1  Cloud Identification

Clouds often occur in multiple layers and have variability in density or opacity and cloud top height. Figure 7 shows the shape of the laser pulses transmitted and those backscattered from clouds. The cloud-returned pulse shape varied



with cloud type and structure. For the analysis used here, the data processing of cloud returns is performed in two steps. In the first step, pulse echoes from significant scattering surfaces are identified from the lidar backscatter profiles. For hard (ground) or relatively opaque surfaces (dense cloud tops), as shown in Fig. 7, the echo width is limited to a 150-m corresponding to the laser's 1 µs pulse width. For signals backscattered from diffuse clouds, we
first subdivided the backscatter profiles into 500-m atmospheric layers. We then labeled those with sufficient backscatter as a pulse echo. The range to each echo was then calculated using the centroid of the backscatter from that layer, as illustrated in Fig.2. In the second step, the cloud echoes were grouped and stratified for every 500-m layers and then aggregated and averaged over 10-s ground track. The averaged line shapes were used to retrieve $XCO_2$ to the averaged centroid cloud height.

The altitude of a significant scattering surface can usually be determined using lidar range, the aircraft GPS altitude and pitch and roll angles. However, during aircraft rolls and turns, distinguishing the altitude of cloud tops from the ground sometimes required using the simultaneous aircraft radar data that provided the nadir range to the ground through the clouds.

### 3.2 Data Processing

There are several steps used in the lidar's retrieval process of $XCO_2$. The data from the $CO_2$ Sounder are calibrated before $XCO_2$ is retrieved by using a line-fitting retrieval algorithm (Abshire et al., 2014). The calibration utilizes a laser energy vs. wavelength correction (<10%), a correction for the transmissions of the receiver's optical band-pass filter (<2%) and, for these flights, a detector nonlinearity correction (<2%). The laser wavelengths are benchmarked in the lab and field by using auxiliary equipment and measurements. The $CO_2$ Sounder pulse energy monitor is
calibrated while the instrument is operating in the field. The outgoing laser pulse energies are monitored using a beam pick-off, integrating sphere and detector. The acquisition of outgoing pulse energy uses the same digitizer as the lidar backscatter. Additional post-flight calibration is made using a flight segment during the engineering flight with known atmospheric conditions and a high resolution $CO_2$ mixing ratio profile measured by an in situ sensor, where parameters are calibrated against atmospheric radiative transfer calculations. This allows assessing the
corrections for detector nonlinearity and the receiver's optical band-pass filter. These calibrations are then applied to all retrievals for the science flights.

   In the forward calculations, we used the spectroscopy database HITRAN 2008 (Rothman et al., 2009) and the Line-By-Line Radiative Transfer Model (LBLRTM; Clough et al., 1992; Clough and Iacono, 1995) V12.1 to calculate $CO_2$ optical depth and create Look-Up-Tables (LUT) for a uniform concentration of 400 ppm. We then use these
LUTs to retrieve the best-fit $CO_2$ concentration by comparing the measured line shape samples with calculated absorption line shapes. The retrievals used atmosphere (pressure, temperature and water vapor profiles) from the near real time forward processing data of the Goddard Earth Observing System Model, Version 5 (GEOS-5; Rienecker et al., 2008). Data on the full model grid ($0.25^o$ latitude x $0.3125^o$ longitude x72 vertical layers, every 3 hours) were interpolated to flight ground track position and time for the atmospheric $CO_2$ absorption calculations.
Absorption line fitting was performed in optical depth with a linear least-square fitting approach. The fitting residuals were spectrally weighed by the square of estimated SNR at each measurement wavelength based on our lidar noise model, which gives more weighing to measurements on line wings than those on line center. The retrieval algorithm solves for Doppler shift, baseline offset, slope, surface reflectance, column-averaged $CO_2$ and $H_2O$ ($XCO_2$ and $XH_2O$) simultaneously for the best fitting. Details of forward calculations and retrieval algorithm
are given in Abshire et al. (2014).

   There is a weak isotopic water vapor (HDO) line centered at 1572.253 nm on the shoulder of the 1572.335 nm $CO_2$ line. Depending on atmospheric water vapor content, this can distort the $CO_2$ line shape and impact the value of the $XCO_2$ retrieval. The $CO_2$ Sounder's wavelength assignments place 1 or 2 laser wavelengths on the HDO line peak. This allows the retrievals to also solve for $XH_2O$, which is important because atmospheric water vapor content is
highly variable in space and time. Passive remote sensing of greenhouse gases, e.g.., OCO-2, GOSAT and TCCON,


measures $O_2$ absorption for column dry air abundance. Measuring water vapor is an alternative way to adjust water vapor data from weather forecast models for better estimate of greenhouse gas mixing ratio. This approach has been recommended in the white paper report of NASA's ASCENDS mission (Jucks et al., 2015, http://cce.nasa.gov/ascends_2015/index.html).

## 4. XCO₂ Measurements to Cloud Tops

During the ASCENDS airborne campaigns in the summers of 2011 and 2014 and the winter of 2013, the $CO_2$ Sounder made measurements to cloud tops over the U.S. west and mid-west. Retrievals of partial column $XCO_2$ were made over low-level marine stratus clouds, cumulus clouds at the top of PBL with cumulonimbus during

thunderstorms, mid-level altocumulus and to visually thin cirrus clouds.

### 4.1 XCO₂ Measurements to the Tops of Marine Stratus Cloud

Marine stratus clouds exist over large portion of ocean adjacent to the west side of continents where ocean currents are cold and a temperature inversion layer is formed to condense the upward-moving moist air. Marine stratus clouds are sheet-like clouds with nearly horizontally uniform base and top and shallow in depth. Once formed, they

may be advected by the wind over land areas. The 2011 ASCENDS airborne campaign had one flight over the Pacific Ocean, west of California coastline on Aug. 2 and flew over marine stratus cloud decks (shown in the left of Fig. 8) with a cloud top elevation of approximately 700-m. The campaign also utilized the AVOCET gas analyzer (Vay et al., 2011) on board for all flights to measure in situ $CO_2$ concentration at every 1-s. During spiral down maneuvers, the AVOCET measured the vertical profile of $CO_2$ concentration. These were used to compare to $XCO_2$

retrievals from the $CO_2$ Sounder lidar. The spiral down segments typically lasted about 30 minutes.

The retrieval results are shown in Figure 8. The right panel shows that the partial column $XCO_2$ retrievals based on a 10-s average have 2~3 ppm standard deviation with biases less than 1 ppm over all flight altitudes, compared to the in situ data from the AVOCET. The retrievals with highest precision were from flight altitude of 8-10 km, indicating the optimal operating altitude for the lidar. At higher altitudes there were fewer returned laser photons and signals

noisier, while at lower altitudes, the path lengths were shorter and absorption signals weaker. Overall, the retrievals results are comparable in quality to those from other 2011 flights under clear conditions (Abshire et al., 2014).

### 4.2 XCO₂ Measurements to the Tops of Cumulus Cloud

Cumulus clouds form as water vapor condenses in a strong, upward air current above the Earth's surface. Cumulus clouds are often seen over land in the afternoon summertime after land surface is fully heated by the Sun. Cumulus

clouds usually have flat bases but lumpy tops. Cumulus clouds grow upward and can develop into a tower-like cumulonimbus, which is a thunderstorm cloud.

The August 10[th] 2011 flight of the ASCENDS airborne campaign flew to Iowa near the West Branch Iowa (WBI) tall tower. The flight passed over many isolated cumulus clouds in the area with cloud tops ranging from 1950-m to 2200-m near the top of PBL. Analysis of pulse echoes from both the cloud tops and the ground within the 100-s data

averaging time allows solving for the partial column $XCO_2$ in PBL by using the differential absorption line shape (Fig. 2). The results showed a strong seasonal drawdown over cornfield in the area and were consistent with the in situ AVOCET data (Ramanathan et al., 2015).

For this work, we performed $XCO_2$ retrievals to the puffy cloud tops for the same flight but use 10-s averaged data. Retrievals made to the cumulus cloud tops near the spiral down segment at the West Branch tall tower had standard

deviations of 3-6 ppm with average biases less than 1 ppm (Fig. 9, left panel). These statistics are based on retrievals within the spiral down flight segment with limited sample size, depending on cloud conditions.





Retrievals of $XCO_2$ to the ground in the same segment showed results standard deviations of 2~4 ppm, and with similar biases, shown in the right panel of Fig. 9. A significant decrease in $XCO_2$ was evident at lower flight altitudes, caused by the large $CO_2$ drawdown in the boundary layer above the cornfield. In this region, the cumulus clouds act as a divider to separate free troposphere $CO_2$ from the boundary layer $CO_2$ that are involved in different

physical processes. The difference between the two $XCO_2$ amounts allows better estimate of surface sources and sinks (Ramanathan et al., 2015).

During the ASCENDS sunset flight from California to Iowa and back on Aug. 25, 2014, there were many cumulus clouds (Fig. 6). Fig. 10 illustrates the detected boundary layer clouds with cloud tops below 2-km and the mid-level clouds with cloud tops around 4-km above ground. In the middle of the flight, a cold front moved through the area

and cumulonimbus clouds were developed vertically and a thunderstorm was formed in the region. The cloud top heights ranged from 2-km for PBL cumulus to as high as 3.5 km for cumulonimbus clouds and the standard deviation of cloud top height was more than 100 m as shown in Fig. 5. The measurement analysis showed the average cloud reflectance was about 6%, which is sufficient to clearly show the gas absorption features across the measurement line and enable quality retrievals of partial column $XCO_2$ to the cloud tops.

We show two segments during the flight to illustrate how $XCO_2$ retrievals to cloud tops may be used to help resolve horizontal and vertical gradients of atmospheric $CO_2$ concentration. Both segments have longer than 5-minute continuous cloud covers and have more than 30 retrievals to cloud tops for statistics. Segment A, marked in Fig. 10, is a 7-minute long segment (23:42–23:49 UTC) near the WBI tall tower during level flight at an altitude of 5-km, while Segment B is a 30-minute long segment (2:30-3:00 UTC) at the similar altitude on the way back to California

after three flights in a square pattern around the WBI tower. Most of the clouds in Segment A were PBL cumulus clouds that had cloud tops around 2-km above ground with some higher cumulonimbus with cloud tops as high as 3.5-km. In Segment B most clouds were cumulus with slightly lower tops around 1.5-km and some were patchy altocumulus clouds with tops around 4-km, which can be clearly seen in Fig. 11. These cloud covers and types can be also identified from photos taken by a digital camera onboard.

Over the 7-minute Segment A with a total of 40 retrievals the lidar measurements of $XCO_2$ to PBL cumulus cloud tops over a 10-s average had negligible bias (395.2 ppm vs. 395.4 ppm of AVOCET) and a standard deviation of 1.94 ppm. The XCO2 retrievals to the PBL cumulus cloud tops from the lidar measurements in the 30-minute long Segment B had a standard deviation of 1.85 ppm from a total of 114 retrievals and a mean value of 393 ppm, which is 2 ppm lower than those in Segment A.

The central location of Segment B is about 250 km west of Segment A. Unfortunately there was no in situ vertical profile data to validate this significant gradient. In this situation, $CO_2$ concentration simulations from the Parameterized Chemical Transport Model (PCTM; Kawa et al., 2004, 2010) were used for inter-comparison. PCTM $CO_2$ concentration simulation is driven by meteorological data from the Modern-Era Retrospective analysis for Research and Applications (MERRA) (Bosilovich, 2013), which is a NASA reanalysis using GEOS-5. The vertical

mixing profile in PCTM is parameterized for both turbulence diffusion in the boundary layer and convection. PCTM in this case is run at $1.25^o$ longitude x $1.0^o$ latitude with 56 hybrid vertical levels and outputs hourly, which should be sufficient to resolve the gradient between these two locations which are $2.4^o$ longitude away and measurements are 3 hours apart.

Figure 12 shows the vertical profile of model $CO_2$ for both segments. Segment A had high $CO_2$ concentration in the

lower atmosphere. We infer this was likely due to the mixing during the thunderstorm and subsequent surface emission in the evening. The vertical profile in Segment B shows a typical summer nighttime vertical structure of $CO_2$ concentration in the area with overall low value in the lower atmosphere after daytime uptake by growing vegetation but high values near the surface when surface uptake stops and respiration starts. The difference in lidar measurements of $XCO_2$ to cloud tops by the lidar between Segment A and B reflects these two different processes

and is consistent with PCTM model simulations. Our $XCO_2$ retrievals to mid-level cloud tops in the middle of




Segment B (2:38-2:48 UTC) were back to high value of 395.8 ppm, on average of 39 retrievals, which excludes the lower $CO_2$ concentration below clouds. During the same 10-minute portion of the segment, our lidar measurements of XCO$_2$ to PBL cumulus cloud tops stayed at 393 ppm, averaged over 28 retrievals. We had 32 clear sky full column $XCO_2$ retrievals to the ground between the popcorn clouds during the 30-minute segment. The average value of full column $XCO_2$ was 389 ppm, which is about 4 ppm lower than $XCO_2$ to cumulus cloud tops and 7 ppm lower than that to the mid-level cloud tops, as illustrated in Fig. 12. The $XCO_2$ measurements to the land surface had a standard deviation of 1.6 ppm, which, as expected, was less than those to the cloud tops. In this case, the lidar measurements of $XCO_2$ to cloud tops allow distinguishing both horizontal and vertical gradient of atmospheric $CO_2$ concentration.

### 4.3 XCO$_2$ Measurements to Cirrus Clouds

Cirrus are thin and semi-transparent clouds, and are globally widespread in the upper troposphere. Cirrus cloud height decreases with latitude, following the tropopause height, and can be as low as 6~8-km in the high latitudes and as high as 16~18-km in the tropics (Sassen et al., 2008). The occurrence frequency of cirrus clouds is about 17% (Sassen et al., 2008) on a global average, but it can be as high as 70% (Nazaryan et al., 2008) in the equatorial west-central Pacific Ocean associated with deep convections at the Intertropical Convergence Zone (ITCZ) and seasonal monsoon circulations. Cirrus clouds are composed of ice crystals and strongly absorptive in our $CO_2$ measurement line (Wiscombe and Warren, 1980; Warren, 1984; Gosse et al., 1995). Therefore, the laser backscatter from cirrus clouds is expected to be substantially lower than clouds composed of water droplets. The reflectance of cirrus clouds varies with cloud physical and spatial structure.

Some cirrus clouds encountered during the ASCENDS airborne campaigns were dense and thick and had sufficient echo pulse energy to show clear $CO_2$ absorption line shape. However, for most cases, the energy values were lower and the absorption line shapes are not sufficiently clear to allow quality retrievals. Fig. 13 show an example of $XCO_2$ retrievals to cirrus cloud tops near the spiral down flight segment in Iowa on March 7, 2013. The data are averaged over 100-s and show a clear $CO_2$ absorption line shape. The aircraft altitude was 12.1 km and the averaged cirrus cloud top height was 10.5 km. The lidar measurements show a retrieval of $XCO_2$ of 392.8 ppm to the cirrus cloud tops, which is lower than the full column $XCO_2$ to the ground of 398 ppm. The lidar retrieval is consistent with in situ AVOCET data of 392.4 ppm for the same layer average in the stratosphere. Unfortunately, there were not enough cases with suitable cirrus cloud tops during these three campaigns to allow calculating statistics.

## 5. Discussion and Conclusion

The pulsed multi-wavelength IPDA lidar approach allows accurately determining the photon path lengths and retrieving $XCO_2$ to cloud tops. Measurements to cloud tops and ground were made with the $CO_2$ Sounder lidar during the 2011, 2013 and 2014 ASCENDS airborne campaigns. These measurements were used to study the $XCO_2$ retrievals made to a variety of cloud tops and to demonstrate the value of these retrievals in resolving both horizontal and vertical gradients of atmospheric $CO_2$. Measurements were made over a variety of clouds, including cumulus and marine stratus at the top of the boundary layer, mid-level altocumulus and cirrus. For all clouds except cirrus, the data processing rate was greater than 90%, excluding cases when aircraft was too close to cloud tops (< 1 km) and when aircraft was tilted substantially (> 10 degree off-nadir).

Analysis of the airborne campaign measurements showed that the laser pulse energies from the tops of boundary layer clouds such as stratus and cumulus were usually sufficient to allow clear identification of $CO_2$ absorption line shape and good retrievals of partial column $XCO_2$ to cloud tops. On average, the reflectance of the boundary layer cloud tops was 5%. In most cases, the boundary layer clouds are too thick for laser pulse to penetrate and allow ground echoes to return simultaneously. However, when over broken clouds, after averaging over 10-s of ground





track or longer, both cloud and ground returns were available. If clouds are patched or broken, retrievals of both $XCO_2$ to the ground and to cloud tops simultaneously and the difference between the two could be then used to estimate the residual $XCO_2$ in the boundary layer, whose value is the most sensitive to surface carbon sources and sinks.

For passive remote sensing approaches, cirrus clouds can significantly modify photon's path length and cause a significant error in $XCO_2$ retrievals. In contrast the lidar measurements showed range resolved pulse echoes from semi-transparent cirrus clouds and ground. In some cases those could be used to retrieve full column $XCO_2$ to the ground and partial column $XCO_2$ to cirrus and then to estimate tropospheric column $XCO_2$. However, in most cases during the campaigns, the backscattered pulse energies from cirrus clouds were low, compared to other clouds such
as stratus and cumulus clouds. Only dense and thick (> 1.0 km) cirrus clouds allowed detection of clear $CO_2$ absorption line shapes and thus yield good $XCO_2$ retrievals. One limitation for these initial airborne measurements was that cirrus clouds were at high-altitude (~10-km) so that the column path length from aircraft altitudes to the cloud tops was short and the $CO_2$ absorption signal was weak. For future space-based missions, the path length of pulse echoes from cirrus clouds will be longer and the $CO_2$ absorption will be stronger, improving retrievals.

The quality of $XCO_2$ retrievals is being improved with advancing technologies for the active remote sensing approach, including both laser and detector toward the measurement goals of ASCENDS. Our results show that $XCO_2$ retrievals to the flat marine stratus cloud tops have the same quality of those to sea surface. Meanwhile, the $XCO_2$ retrievals to the puffy cumulus or cumulonimbus cloud tops were degraded by a factor of 1.2, compared to those to the ground. Previous ASCENDS Observing System Simulation Experiments (OSSE) with clear-sky
measurements (Kawa et al., 2010; Hammerling et al., 2015) have shown that lidar approaches have the greater spatial and temporal coverage than passive approaches and hence a higher potential to reduce uncertainties in carbon budget estimates. Retrievals to all-level cloud tops with corresponding measurement precision are planned to be included in future OSSE studies to assess their impact on atmospheric transport modeling and surface flux estimates.

Partial column $XCO_2$ retrievals to different cloud tops and to the ground allow distinguishing horizontal and vertical
gradient of atmospheric $CO_2$. This measurement capability for the future space carbon missions will be particularly valuable for the regions with persistent cloud covers, e.g., tropical ITCZ, west coasts of continents with marine layered clouds and southern ocean with highest occurrence of low-level clouds, where underneath carbon cycles are active but where measurements from passive satellite-based spectrometers are limited. Lidar-based measurements to cloud tops will fill these significant gaps, provide a more complete picture of the $CO_2$ distribution and will benefit
atmospheric transport modeling as well as global and regional carbon budget estimates.

**Acknowledgement:** This work was supported by the NASA ESTO IIP program and the NASA ASCENDS pre-formulation activity. We gratefully acknowledge the work of the DC-8 team at NASA Armstrong Flight Center for helping plan and conduct the flight campaigns.

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



Table 1. Lidar measurements of surface reflectance during the 2013 and 2014 ASCENDS science flights (SF) over a variety of surface types, including ocean, snow and clouds. Reflectance of 0.4 over desert was specified as a standard to quantify reflectance over other surface types.

| Surface | Reflectance | Flight | Measurement Location |
|---|---|---|---|
| Desert | 0.4 (established standard) | 2014 SF2<br>2013 SF1<br>2013 SF2 | Edwards AFB, CA<br>Owens Valley, CA<br>Railroad Valley, NV |
| Semi-desert | 0.32 | 2014 SF2 | Great Basin Range, NV |
| Desert/Cropland | 0.25-0.35 | 2013 SF1<br>2014 SF1 & SF4 | Central Valley, CA<br>Central Valley, CA |
| Cropland (winter) | 0.30 | 2013 SF5 | Great Plains, CO/NE/IA |
| Mountain/Forests | 0.25-0.30 | 2014 SF3<br>2014 SF4 | Rocky Mountains, CO<br>Sierras, CA |
| Cropland (summer) | 0.20 | 2014 SF3 & SF5 | Iowa |
| Forests | 0.15-0.25 | 2014 SF1 | N. California forests |
| Ocean (normal incidence) | 0.10-0.20 | 2014 SF2 | Pacific Ocean |
| Ocean (slant incidence) | 0-0.10 | 2014 SF2 | Pacific Ocean |
| Snow (cold) | 0.05-0.10 | 2013 SF4 | Rockies, CO |
| Snow (warm) | 0.02-0.10 | 2013 SF5 | Midwest, IA/MO |
| Clouds | 0.02-0.10 | 2014 SF1, SF2 & SF3 | West and Midwest |



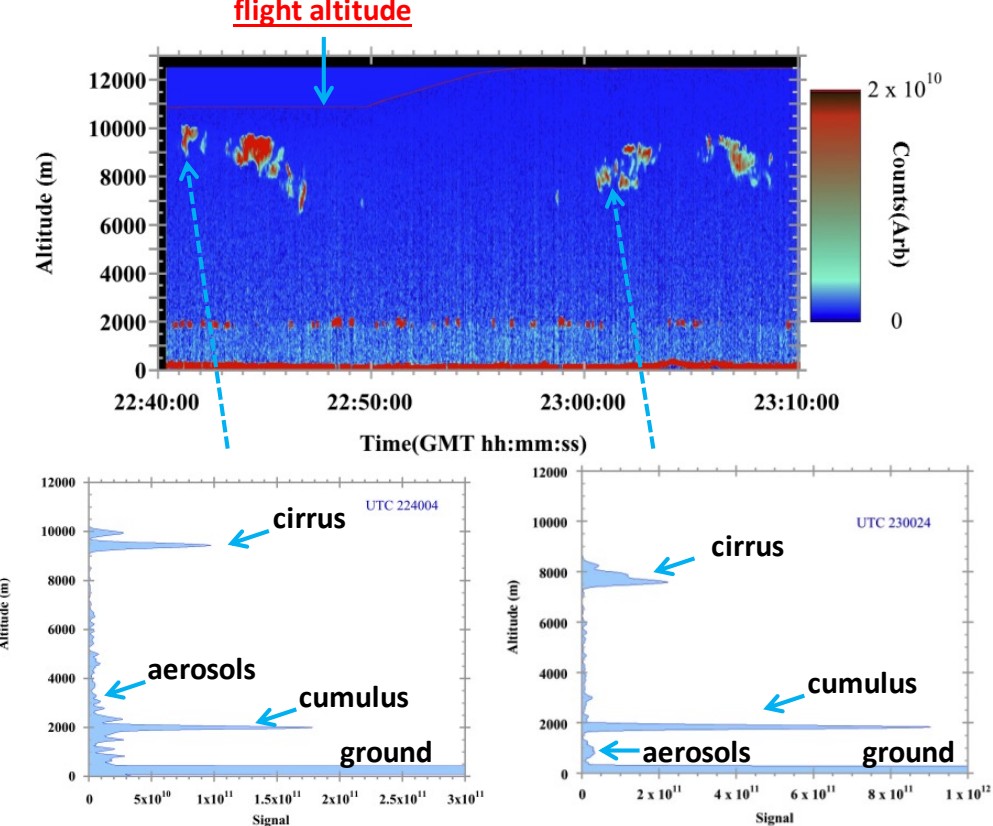

Figure 1. A 30-minute long vertical cross-section (top panel) and two individual 1-s vertical profiles (bottom panels) of atmospheric backscattering at off-line wavelengths of $CO_2$ line measured by NASA GSFC $CO_2$ Sounder near West Branch Iowa tall tower site on Aug. 11, 2011. The backscatter signals were corrected by square of range and averaged by 1 μs vertical running mean (150-m) and 1-s horizontal running mean (200-m of ground track). Returns from ground, cumulus and cirrus clouds, and aerosols are illustrated and labeled.



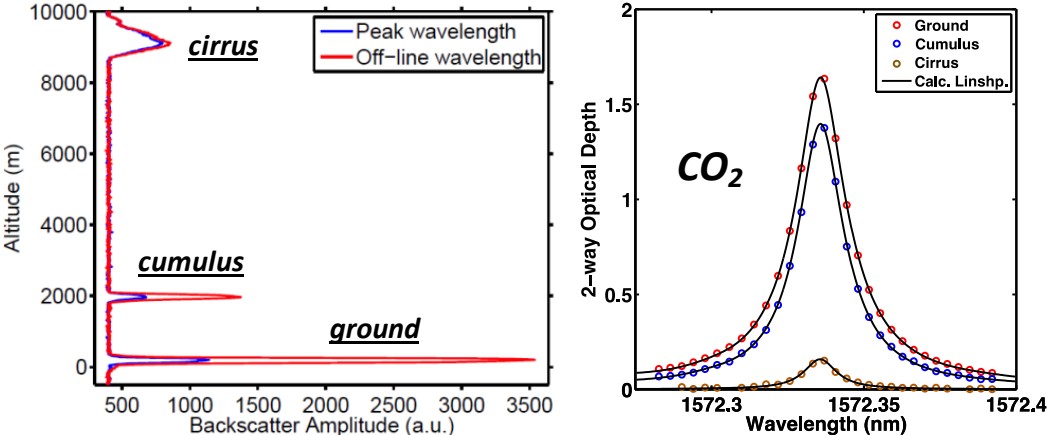

Figure 2. Backscatter profile (left panel) and $CO_2$ absorption line shapes (right panel) in terms of optical depth for laser returns from ground, cumulus clouds and cirrus clouds near West Branch Iowa tall tower site on Aug. 10, 2011. In the left panel, the backscatter profile of off-line is plotted in red, and the backscatter profile at line center with peak absorption is plotted in blue. Both optical depth and differential optical depth between off-line and line center wavelength increases with photon path length or range between airplane and the scattering surface. Data are averaged over 10-s of ground track for both plots. In the right panel the dots are the lidar measurements and the solid lines are the best-fit line shapes from the $XCO_2$ retrievals.



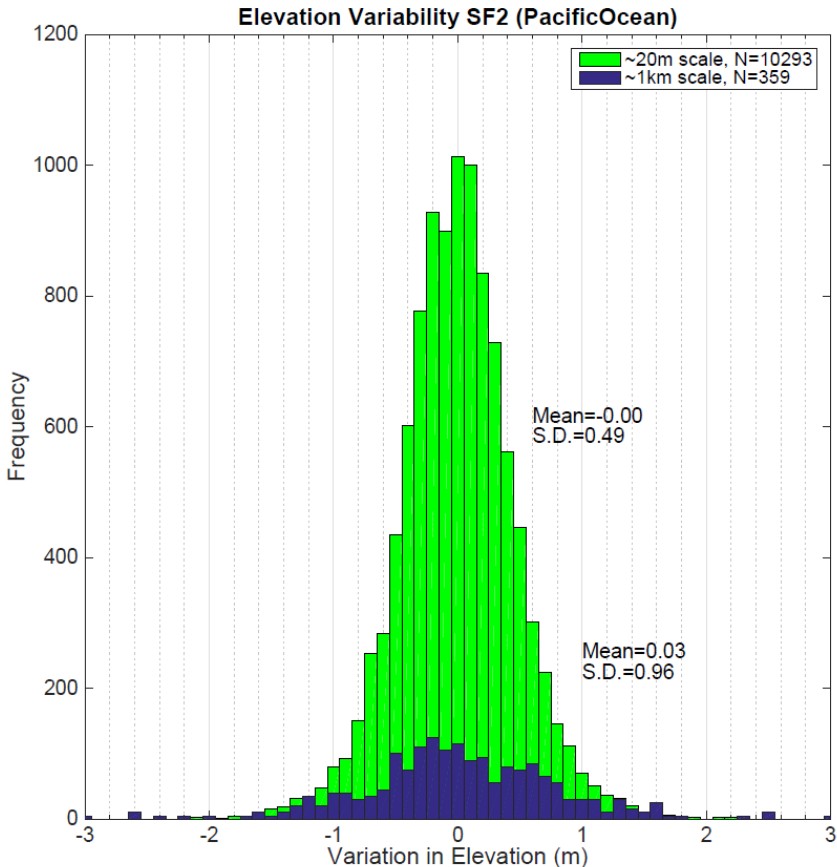

Figure 3. Histogram of the variation in sea surface elevation on the Aug. 22, 2014 ASCENDS flight over Pacific Ocean near California coast. The green bars are raw data for every 0.1-s integration time (20-m scale along track) and the blue bars are averages over 5-second (1-km scale along track).





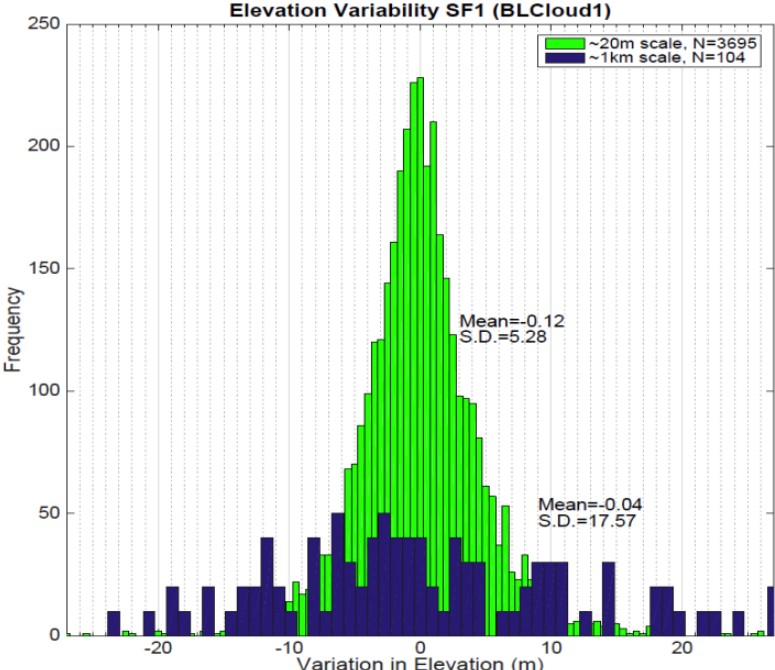

Figure 4. Histogram of the variation in cloud top height after averaged for 0.1-s (in green) and 5-s (in blue), respectively, for the Aug. 20, 2014 flight above stratus clouds along California coastline, showing a 0.1-s standard deviation of 5.2 m and a 5-s standard deviation of 18 m..





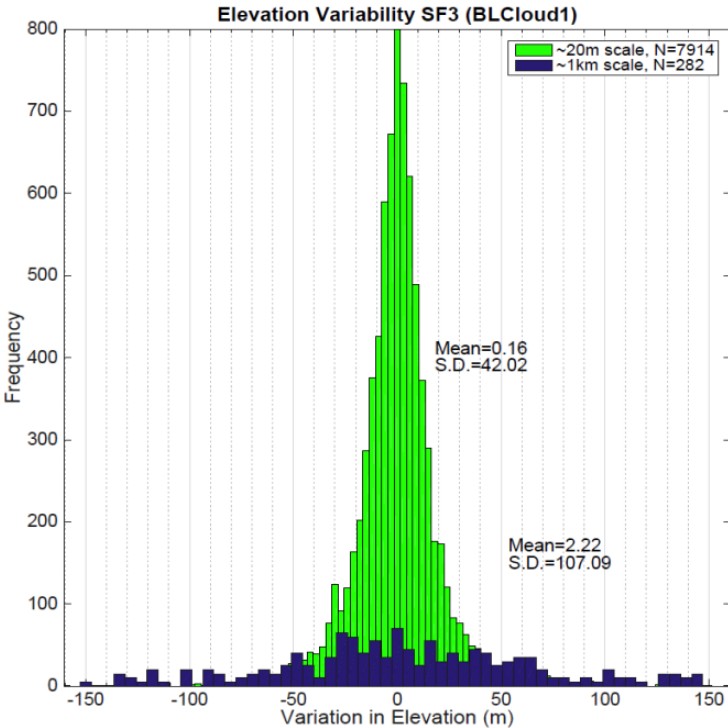

Figure 5. Same as Fig. 4 for the Aug. 25, 2014 flight above cumulus clouds in Iowa, showing a 0.1-s standard deviation of 42 m and a 5-s standard deviation of 107 m.





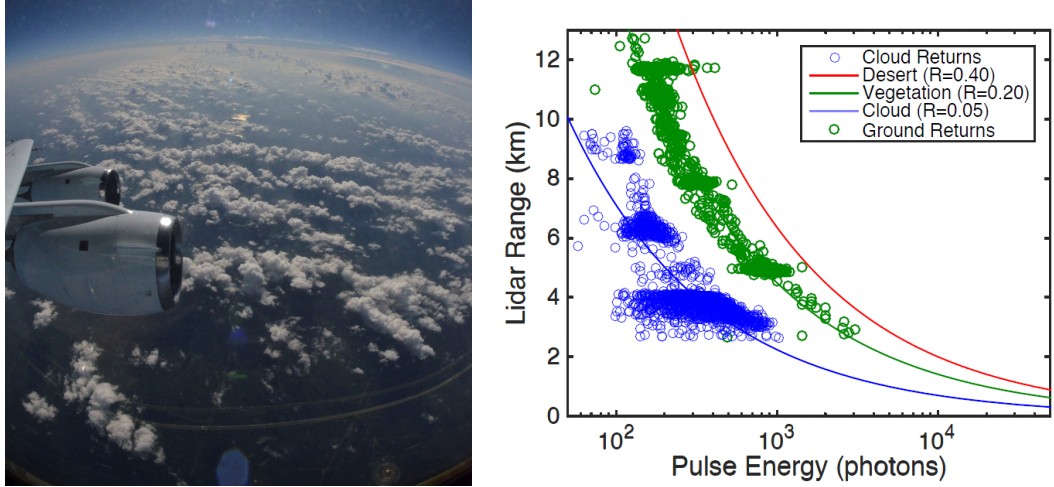

Figure 6. The left photo shows the cumulus clouds above PBL from the ASCENDS sunset flight on Aug. 25, 2014 near the West Branch Iowa tall tower. The right panel shows the returned pulse energy in number of photons as a function of lidar range from airplane altitude for the cumulus clouds. The average ground reflectance (in green) is approximately 20%, while the average cumulus cloud top reflectance is about 5% (in blue) and shows more variability.



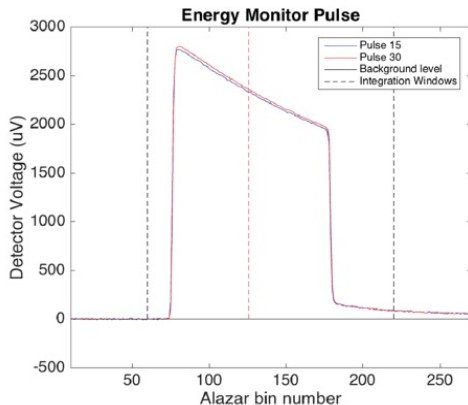
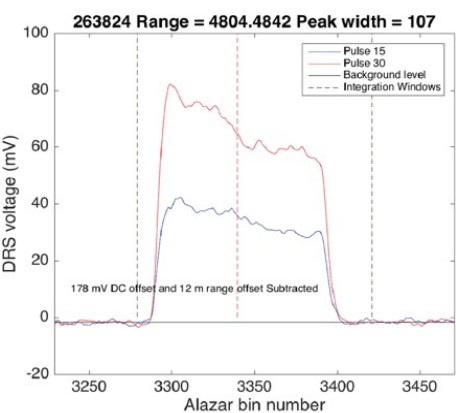

Figure 7. The lidar transmitted pulse shape (left panel) and the recorded echo pulse shape returned from a cloud top (right panel). The blue lines are for pulse #15 that is near $CO_2$ absorption line center and the red lines are for pulse #30 in the line wing.





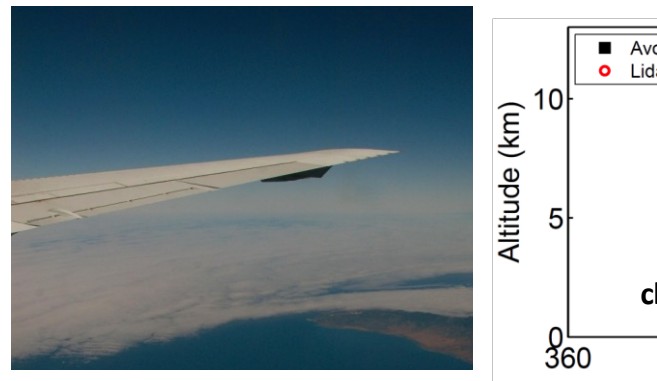
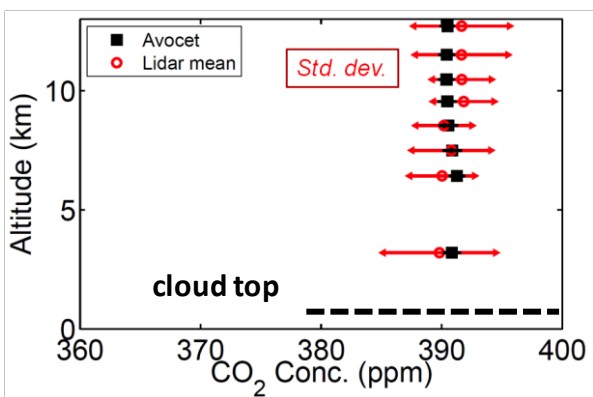

Fig. 8. (Left panel) Photo of marine stratus cloud deck over the Pacific Ocean near the California coastline taken on the ASCENDS flight on Aug. 2, 2011. (Right panel) The retrieved values of $XCO_2$ to the cloud tops at altitudes of 700-m (black dashed line) as a function of flight altitude. The $XCO_2$ values integrated from in situ AVOCET gas analyzer are marked in black squares and the retrieved values from the $CO_2$ Sounder for 10-s average are marked in red circles. The error bars for the retrieved $XCO_2$ are for ±1 standard deviation.





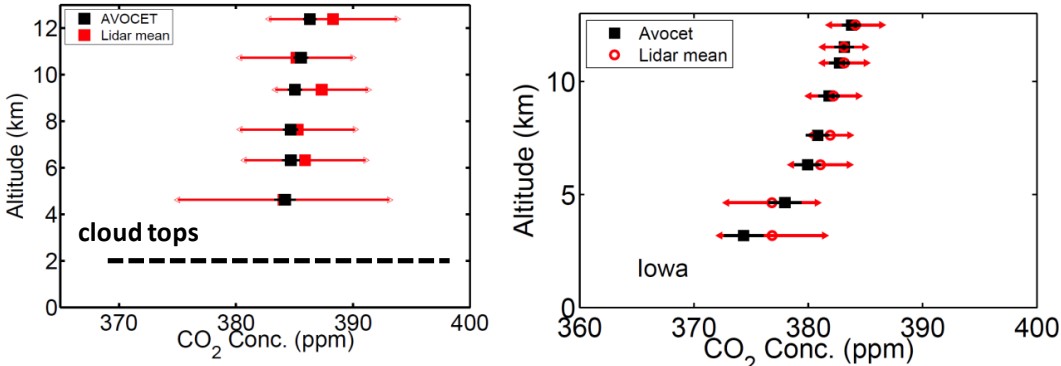

Figure 9. The XCO$_2$ retrievals for lidar measurements to the tops of broken cumulus clouds (left panel) and to the ground (right panel) on the Aug. 10, 2011 ASCENDS flight over Iowa. The XCO$_2$ values from in situ AVOCET gas analyzer are marked in black squares and the values of XCO$_2$ retrievals from CO$_2$ Sounder measurements averaged over 10 seconds are marked in red circles with error bars of ±1 standard deviation. The average altitude of cloud tops (~2 km) is plotted in the dashed line.



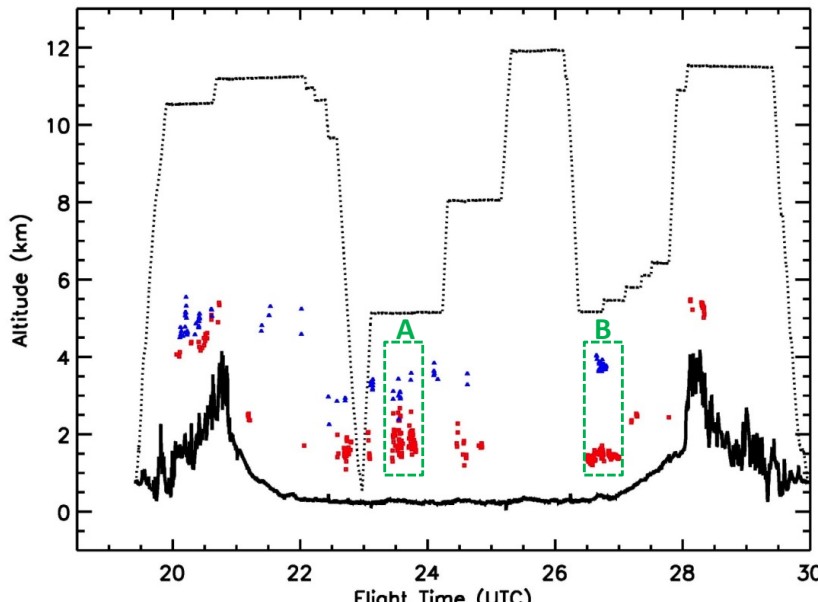

Figure 10. Summary plot of altitude-resolved lidar measurements for the sunset ASCENDS flight to Iowa on Aug. 25, 2014. The aircraft altitude is plotted in the dotted black line, the ground elevation is plotted in the solid black line, the altitudes of boundary layer cloud tops are plotted in the red squares and the altitudes of mid-altitude clouds tops are plotted in the blue triangles. Green boxes 'A' and 'B' are two segments selected for further data analysis.



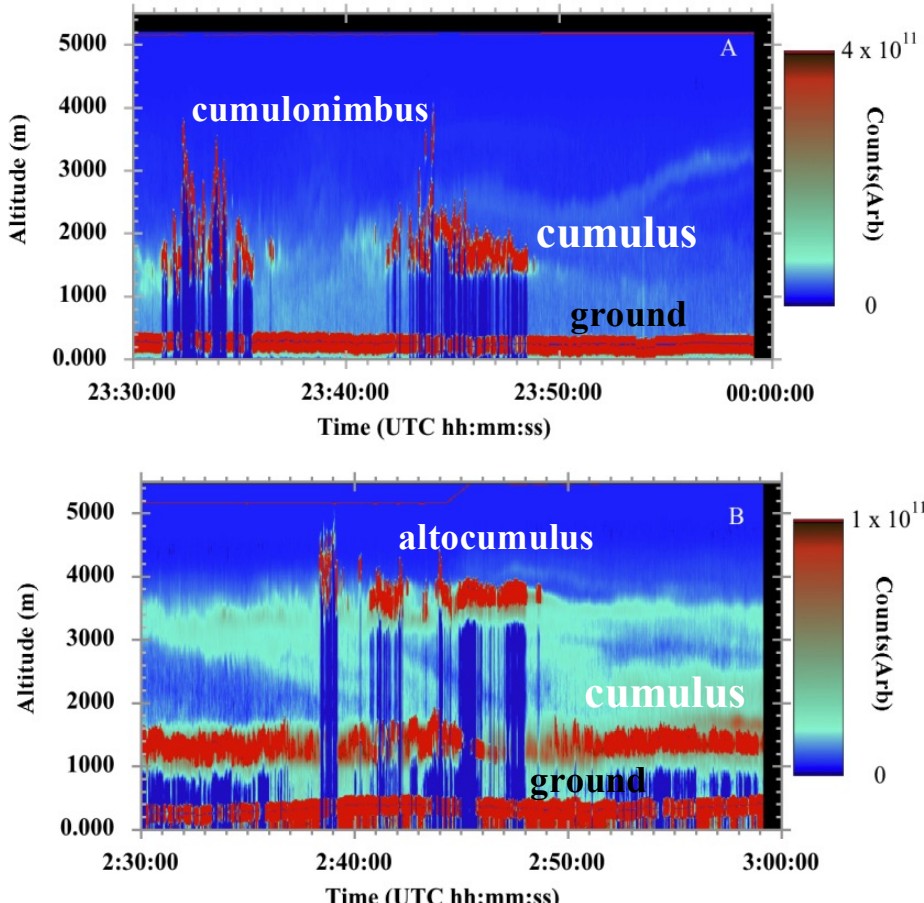

Figure 11. Vertical cross-sections of range-corrected backscattered pulse energy for Segment A and B marked in
Fig. 10. The lidar returns from the ground are at the bottom and cloud returns are at a variety of altitudes from 1 to 4
5    km. The red lines on the top of plots indicate aircraft flight altitudes.





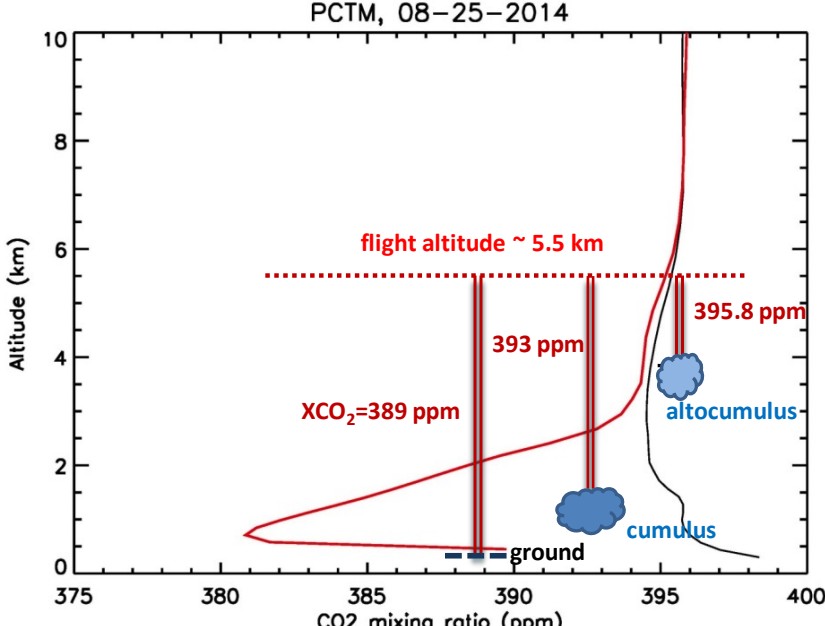

Figure 12. Vertical profiles of $CO_2$ mixing ratio on Aug. 25, 2014 for the central location of Segment A at (41.1°N, 92.3°W) in black and for Segment B at (41.3°N, 94.7°W) in red from the NASA Parameterized Chemical Transport Model. The $XCO_2$ measurements to ground, PBL cumulus clouds and mid-altitude altocumulus clouds from the $CO_2$ Sounder lidar for Segment B are labeled, respectively. Flight altitudes were around 5.5 km above sea level for both segments.



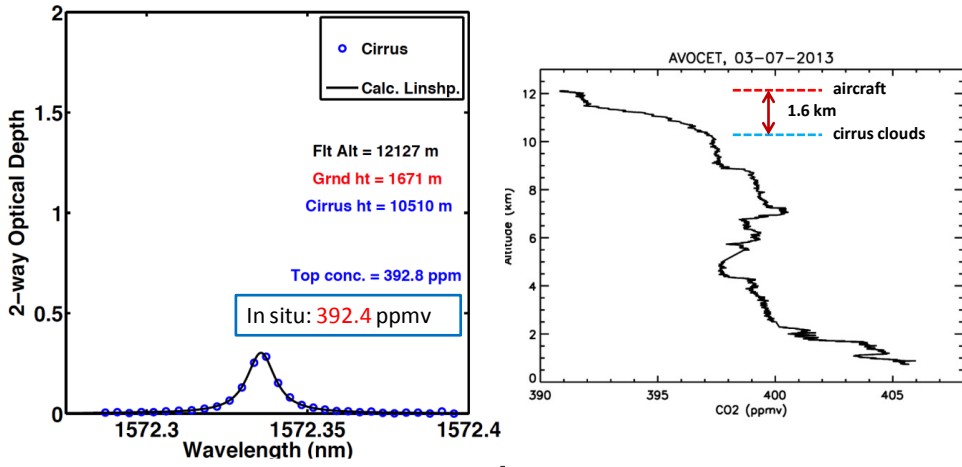

Figure 13. A $CO_2$ absorption line shape measured on the March 7, 2013 to cirrus cloud tops at 10.5 km altitude (left panel). The lidar measurements are the blue circles and the fitted line shape is the solid black line. AVOCET in situ vertical profile of $CO_2$ concentration is plotted at the right panel. The aircraft flight altitude was 12.1 km and the lidar range to cirrus cloud tops was 1.6 km.