# Peer review of "Measurement of Atmospheric CO2 Column Concentrations to Cloud Tops with a Pulsed Multi-wavelength Airborne Lidar"

_Atmospheric Measurement Techniques, 2017_

## Referee Comment (RC1) · Anonymous Referee #1 · 11 Aug 2017

**General Comments**

Jianping Mao et al. report on airborne $CO_2$ measurements using an integrated path differential absorption (IPDA) lidar with focus on cloud tops as the backscatter targets. Since certain cloud coverage is present at most parts of the world, airborne and especially future spaceborne IPDA lidars will be faced with these conditions frequently. The presented specific analysis of IPDA lidar measurements to cloud tops gives valuable information about the performance of the IPDA technique for these situations. The authors show measurement results for different common cloud types and its characteristics. Additionally, the possibility to retrieve partial column mixing ratios – especially for the boundary layer – for cases, at which measurements to cloud tops and the ground are made alternatingly, is shown.

The paper is well written and gives a detailed overview of IPDA measurements to cloud tops. The structure is clear and all information is presented comprehensibly (except for some minor details mentioned below). I recommend the publication in AMT.

**Specific Comments**

Section 2, p. 2, l. 24/25: "This is considerably higher than that of GOSAT…"

I guess, it is meant: "The laser's spectral line-width is considerably narrower than the spectral resolution of GOSAT…". Please reword accordingly.

p. 2, l. 35: "The range backscatter profiles are recorded for all laser wavelengths at a 10 Hz rate."

Why 10 Hz? Does it mean that the raw pulses are accumulated down to a 10 Hz rate to get a better SNR for the range determination? Please describe this fact clearer.

p. 3, l. 23: "The standard deviation increased to about 1 m after measurements are averaged over 5 seconds"

Do you mean the standard deviation of the original 10-Hz range measurements inside averaging intervals of several seconds? This is indicated by the statements in lines 15-20 and seems to be comprehensible. In contrast, in the following it sounds like, if the standard deviations for series of averaged range measurements (inside a certain flight section) increase with longer averaging intervals. This doesn't seem to be logical. If an averaging is done on the scale of the variations or longer, a smoothing should occur. Maybe this is expressed mistakably or there is some information missing about calculation details that can substantiate the statements. See also p. 3, l. 29-35.

Additionally, the impact and the conclusions of these findings should be explained more in detail.

The same as above.

Figure 7, right panel:

Is this a single pulse measurement or averaged?

Section 3.1

How are clouds distinguished from ground (except by using radar data in some cases), e.g. in case of pronounced topography of a flight section, as shown in Figure 10?

Section 4.1, p. 6, l. 18

Was it possible to fly the spiral down to the ground? If not, how were the missing data for the complete column obtained?

Section 4.1, p. 6, l. 19/20

Are any special calculations necessary (e.g. weighting) to get the AVOCET $XCO_2$ data in such a manner that they are directly comparable to the lidar $XCO_2$ data? Or do both represent the same column averaged mixing ratios inherently?

Section 4.2, lines 39ff, and Figure 12

The vertical profiles of the model $CO_2$ are mentioned and shown in the Figure, but the resulting model $XCO_2$ (like above the AVOCET $XCO_2$) is not given here.

Section 5, p. 9, l. 18

Please repeat shortly the reason of the degradation and please add which quantity (instead of "retrieval") is degraded by the given factor.

**Technical Corrections**

15 MHz correspond to 0.0005 cm$^{-1}$.

Should be "Table 1".

---

## Referee Comment (RC2) · Anonymous Referee #2 · 13 Sep 2017

**Measurement of Atmospheric CO2 Column Concentrations to Cloud Tops with a Pulsed Multi-wavelength Airborne Lidar**

**Corresponding Author**: Jianping Mao

**General Comments:**

The authors describe use of a pulsed integrated-path, differential absorption lidar to measure $CO_2$ column abundances between an aircraft and various cloud decks and the ground. Measurements are compared with in situ measurements taken aboard the same aircraft. The "cloud slicing" technique provides advantages relative to passive measurements of $CO_2$ in that $CO_2$ column amounts can be measured in the presence of cloud fields, which is currently not possible with passive instruments. The manuscript is clear, concise, and well-written. I recommend publication with only minor changes and clarifications as described below.

**Specific Comments:**

P. 3, L. 22: In the text discussing figure 2, there is no description of the line shape calculation. From what altitude were the met parameters obtained for use in calculating the line shapes? Were these parameters obtained in an aircraft spiral (which is implied later in the manuscript, but not in this section)? If they were obtained via aircraft spiral maneuver, what is the spatial and temporal separation between this $CO_2$ line retrieval and the sampling spiral?

P. 7, Line 39: I found this paragraph and Figure 12 difficult to follow, and it may need clarification for readers not familiar with the measurement. I suggest clarification in this section that the lidar measurements are column integral measurements, and that the in situ measurements plotted in figure 12 are point measurements at each altitude. For clarification, can the plot include the integral value of the in situ measurement at each altitude that the lidar is being compared with for a more direct comparison? Otherwise, the reader needs to do a mental integral to verify that indeed, the lidar and in situ columns agree at the top of the cumulus clouds (~1.7 km) even though the lidar column value of 393 ppm is very different than the in situ point value of ~385 ppm at that altitude.

P. 19, Figure 6: Why are there no ground returns below the lowest cloud deck on this figure?

P. 22, Figure 9: the $CO_2$ sounder instrument measurements show a steady trend of decreasing $CO_2$ with decreasing altitude, in agreement with in situ measurements, but the $CO_2$ sounder measurements clearly show an end to the drawdown trend in the lowest altitude bin, with the in situ data clearly showing a continuing drawdown. Is there an explanation for the ~4 ppm high bias of the remote measurement relative to the in situ measurement at this altitude?

**Technical corrections:**

P. 1, L. 13: multiple wavelengths should be multi-wavelength

P. 1, L. 34: present should be presence

P. 2, L 3: recommending adding citation for NRC report in references list

P. 2, L 8: suggest adding "and other targets" after cloud tops since ranging applies to the ground measurements as well.

P. 2, L. 33: Should "after passing through" be replaced with "pass through" to be clear that the NB filter is after the receiver telescope and not before it?

P. 3, L. 15: need to add "being" after "before"

P. 4, L. 6: need "the" before "field-of-view"

P. 4, L. 29: by "quality" do the authors mean high precision or high accuracy? What is the definition of quality?

P. 5, L. 5: what is the definition of "sufficient backscatter"? Is there some metric relative to the noise floor that is used as a discriminator?

P. 5, L. 33: Reinecker et al., 2008 is cited in the text but the reference list only includes Reinecker, et al., 2015.

P. 5, L. 36: Should "weighed" be "weighted"?

P. 5. L. 45: please define TCCON.

P. 6, L. 4: recommend adding citation for Jucks, et al., in references list

P. 6, L. 12: suggest changing west side of continents to west side of the United States for clearer context.

P. 6, L. 22: the standard deviations of the two high-altitude and one low-altitude data points appear to be larger than 2-3 ppm as stated, perhaps 2-5 ppm. Can the authors verify this stated range?

P. 6, L. 23: please define AVOCET

P. 6, L. 23: should be "are" after signals

P. 6, L. 40: as stated for line 22 above, the standard deviations of the one low-altitude data point appears to be larger than 3-6 ppm as stated, perhaps 3-10 ppm. Can the authors verify this stated range?

P. 7, L. 1: need "with" after "results"

P. 8, L. 30: I recommend adding some quantitative results to this section such as accuracy and precision values relative to correlated in situ measurements. The conclusion section is mostly a restatement of the introduction without quantitative results.

P. 8, L. 37: that statement about which cases are excluded due to proximity to cloud tops and aircraft tilt would be useful at the beginning of the manuscript, perhaps in section 2.0.

P. 11, L. 23: Ramanathan, et al., 2013 does not appear to be cited in the manuscript text.

Pgs. 16-17: Figures 3 and 4: are there nadir camera images from the flights that could be added to these figures to provide a visual description of the cloud fields? The pictures on figures 6 and 8 are very helpful for describing the cloud field and should be included with figures 3 and 4 if possible.

P. 20, Figure 7: Is the range value in meters here? Suggest adding the unit. What is the definition of the red dotted line (not defined in the legend)? I suggest using thicker lines in the plot for increased readability, and adding the word "dense" before "cloud top" in the caption.

P. 21, Figure 8: AVOCET should be all caps in the legend. In line 2 of the caption, do the authors mean "altitude intervals" instead of "attitudes"?

P. 25, Figure 12: suggest adding "(shown as red dashed line)" at the end of the caption to define the red dashed altitude line in the plot.

---

## Author Comment (AC1) · 27 Sep 2017

**From Reviewer #1**

General Comments

Jianping Mao et al. report on airborne CO2 measurements using an integrated path differential absorption (IPDA) lidar with focus on cloud tops as the backscatter targets. Since certain cloud coverage is present at most parts of the world, airborne and especially future spaceborne IPDA lidars will be faced with these conditions frequently. The presented specific analysis of IPDA lidar measurements to cloud tops gives valuable information about the performance of the IPDA technique for these situations. The authors show measurement results for different common cloud types and its characteristics. Additionally, the possibility to retrieve partial column mixing ratios – especially for the boundary layer – for cases, at which measurements to cloud tops and the ground are made alternatingly, is shown. The paper is well written and gives a detailed overview of IPDA measurements to cloud tops. The structure is clear and all information is presented comprehensibly (except for some minor details mentioned below). I recommend the publication in AMT.

>>> Thanks for your careful review and for your constructive comments that will improve the paper. We also appreciate reviewer's vision about how valuable these measurements to cloud tops are for the future space mission since most of measurements from space are partially or fully covered by clouds. By adding measurements to cloud tops, our IPDA technique enhances global coverage of atmospheric XCO2 and provides some information of its vertical distribution.

Specific Comments

Section 2, p. 2, l. 24/25: "This is considerably higher than that of GOSAT…" I guess, it is meant: "The laser's spectral line-width is considerably narrower than the spectral resolution of GOSAT…". Please reword accordingly.

>>> Reworded. Yes, the narrow line-width of laser means higher spectral resolution.

p. 2, l. 35: "The range backscatter profiles are recorded for all laser wavelengths at a 10 Hz rate." Why 10 Hz? Does it mean that the raw pulses are accumulated down to a 10 Hz rate to get a better SNR for the range determination? Please describe this fact clearer.

>>> As suggested, we revised this statement to "The range backscatter profiles are accumulated and recorded after averaging for all laser wavelengths at a 10 Hz rate to improve single-to-noise ratio (SNR)."

p. 3, l. 23: "The standard deviation increased to about 1 m after measurements are averaged over 5 seconds" Do you mean the standard deviation of the original 10-Hz range measurements inside averaging intervals of several seconds? This is indicated by the statements in lines 15-20 and seems to

be comprehensible. In contrast, in the following it sounds like, if the standard deviations for series of averaged range measurements (inside a certain flight section) increase with longer averaging intervals. This doesn't seem to be logical. If an averaging is done on the scale of the variations or longer, a smoothing should occur. Maybe this is expressed mistakably or there is some information missing about calculation details that can substantiate the statements. See also p. 3, l. 29-35.

Additionally, the impact and the conclusions of these findings should be explained more in detail.

>>> A very good catch. We described the scattering surface roughness as the relative roughness with respect to the local neighborhood, which is the surface elevation change from one point to next. So that is why in the 5-s averaged data the elevation change from one point to next was greater than that in 10 Hz data since here flight time is equivalent to horizontal flight distance. We added some details about that as following,

"The elevations of cloud tops can vary significantly. Lidar measurements showed the standard deviation of marine stratus cloud top heights from the 2014 flights at the California coastline was approximately 5 m for a 0.1-s averaging time and increased to 18 m for 5-s averages, as shown in Fig. 4, which is reasonably consistent with estimates from the 2011 flights over the Pacific Ocean (Abshire et al., 2013). As expected the range measurements to puffy popcorn-like cumulus cloud tops made in the 2014 campaign showed more variation. The standard deviation of the relative cumulus cloud top height changes from one point to next was 42 m for 0.1-s averages and 107 m for 5-s averages, as shown in Fig. 5. Thus, the partial column XCO2 measurements made to cumulus cloud tops using 10-s averaged data are expected to be noisier than these over marine stratus clouds."

Figures 3 – 5. The same as above.

>>> changed

Figure 7, right panel: Is this a single pulse measurement or averaged?

>>> This is a single pulse measurement.

Section 3.1

How are clouds distinguished from ground (except by using radar data in some cases), e.g., in case of pronounced topography of a flight section, as shown in Figure 10?

>>> A good question. We calculated scattering surface elevations from the lidar and compared those to elevations of the ground from the radar data to separate clouds from ground for all cases.

Section 4.1, p. 6, l. 18

Was it possible to fly the spiral down to the ground? If not, how were the missing data for the complete column obtained?

>>> DC-8 usually spirals down over a local airport and can fly horizontally as low as 50-m above airport runway. So the data in the bottom 50-m data was extrapolated to the surface for column average.

Section 4.1, p. 6, l. 19/20

Are any special calculations necessary (e.g. weighting) to get the AVOCET $XCO_2$ data in such a manner that they are directly comparable to the lidar $XCO_2$ data? Or do both represent the same column averaged mixing ratios inherently?

>>> We use the vertical averaging kernel from lidar $XCO_2$ retrieval to compute AVOCET $XCO_2$ with its vertical profile for comparison between the two $XCO_2$ values as column averaged mixing ratios.

Section 4.2, lines 39ff, and Figure 12

The vertical profiles of the model $CO_2$ are mentioned and shown in the Figure, but the resulting model $XCO_2$ (like above the AVOCET $XCO_2$) is not given here.

>>> We used model vertical profiles here as a reference to show the two distinguished vertical structures and see if lidar $XCO_2$ can well represent these vertical and horizontal (between Segment A and B) gradients. We used AVOCET but not model data for lidar $XCO_2$ validation like for Segment A.

Section 5, p. 9, l. 18

Please repeat shortly the reason of the degradation and please add which quantity (instead of

"retrieval") is degraded by the given factor.

>>> We repeated the reasons of the degradation here as suggested, "due to the larger cloud top roughness as well as the lower cloud reflectivity at the measurement wavelengths"

Technical Corrections

p. 2, l. 24

15 MHz correspond to 0.0005 $cm^{-1}$.

>>> Corrected. Thanks.

p. 4, l. 31:

Should be "Table 1".

>>> Corrected. Thank you.

**From Reviewer #2**

General Comments:

The authors describe use of a pulsed integrated-path, differential absorption lidar to measure $CO_2$ column abundances between an aircraft and various cloud decks and the ground. Measurements are compared with in situ measurements taken aboard the same aircraft. The "cloud slicing" technique provides advantages relative to passive measurements of $CO_2$ in that $CO_2$ column amounts can be measured in the presence of cloud fields, which is currently not possible with passive instruments. The manuscript is clear, concise, and well-written. I recommend publication with only minor changes and clarifications as described below.

>>> We appreciate the reviewer's careful reading, corrections and comments which made presentation and interpretation of results better and the statements more precise.

Specific Comments:

P. 3, L. 22: In the text discussing figure 2, there is no description of the line shape calculation. From what altitude were the met parameters obtained for use in calculating the line shapes? Were these parameters obtained in an aircraft spiral (which is implied later in the manuscript, but not in this section)? If they were obtained via aircraft spiral maneuver, what is the spatial and temporal separation between this $CO_2$ line retrieval and the sampling spiral?

>>> Figure 2 is an illustration of measured backscatter profiles and calculated $CO_2$ absorption line shapes from different significant scattering surfaces. We described the forward calculations later in the Section 3.2. We used meteorological data from model that were interpolated to the flight track and time. So the measured and calculated are co-located and simultaneous for the best retrievals.

P. 7, Line 39: I found this paragraph and Figure 12 difficult to follow, and it may need clarification for readers not familiar with the measurement. I suggest clarification in this section that the lidar measurements are column integral measurements, and that the in situ measurements plotted in figure 12 are point measurements at each altitude. For clarification, can the plot include the integral value of the in situ measurement at each altitude that the lidar is being compared with for a more direct comparison? Otherwise, the reader needs to do a mental integral to verify that indeed, the lidar and in situ columns agree at the top of the cumulus clouds (~1.7 km) even though the lidar column value of 393 ppm is very different than the in situ point value of ~385 ppm at that altitude.

>>> Thanks for your suggestion and regret the confusion. The purpose of the plot is to demonstrate how XCO2 retrievals respond to horizontal and vertical gradients of CO2. Unfortunately, we didn't have in situ CO2 data for Segment B for retrieval validation. So we decided to use model data as a reference for both segments for inter-comparison. Both vertical profiles in the plot are from model, not from in situ data. Meanwhile, we do have another version of the figure attached to include the in situ profile in Segment A which didn't show significant vertical gradient.

[Figure]

P. 19, Figure 6: Why are there no ground returns below the lowest cloud deck on this figure?

>>> It is because the cumulus clouds are optically dense and the laser can't penetrate to the ground for a single pulse. Hence there were no ground returns recorded. We interpreted that in the 2nd paragraph of Section 5.

P. 22, Figure 9: the CO2 sounder instrument measurements show a steady trend of decreasing CO2 with decreasing altitude, in agreement with in situ measurements, but the CO2 sounder measurements clearly show an end to the drawdown trend in the lowest altitude bin, with the in situ data clearly showing a continuing drawdown. Is there an explanation for the ~4 ppm high bias of the remote measurement relative to the in situ measurement at this altitude?

>>> This is a good catch and question. We think there are two reasons to cause noisy XCO2 retrievals at the lowest altitude. First reason is the shorter path-length at lower altitudes which gives smaller CO2 absorption signal or OD. The second reason is because we turned off laser once flight altitude was below 3000 ft above ground as required by FAA and turned it back when flight altitude was above 3000 ft above ground. We usually had to turn the laser off before the 3000 ft altitude FAA limit because the increase of detector signal can start causing the lidar detector to get saturated. When we climbed out after a spiral down, the aircraft had larger pitch angles and the laser had larger off-nadir viewing angles and needed some time to settle after being turned back on. These are the reasons why we had noisier data and lower quality retrievals at lower altitudes. In addition, we had fewer data points for statistics in the lower atmosphere. We need to do more tests to set better criteria for retrieval.

Technical corrections:

P. 1, L. 13: multiple wavelengths should be multi-wavelength

>>> Corrected.

P. 1, L. 34: present should be presence

>>> Corrected. Thank you.

P. 2, L 3: recommending adding citation for NRC report in references list

>>> Added

P. 2, L 8: suggest adding "and other targets" after cloud tops since ranging applies to the ground measurements as well.

>>> Added

P. 2, L 33: Should "after passing through" be replaced with "pass through" to be clear that the NB filter is after the receiver telescope and not before it?

>>> Revised

P. 3, L. 15: need to add "being" after "before"

>>> Added

P. 4, L. 6: need "the" before "field-of-view"

>>> Added

P. 4, L. 29: by "quality" do the authors mean high precision or high accuracy? What is the definition of quality?

>>> The 'quality' here means both high precision and high accuracy

P. 5, L. 5: what is the definition of "sufficient backscatter"? Is there some metric relative to the noise floor that is used as a discriminator?

>>> That was compared to the background level. We reworded it to 'significant backscatter'.

P. 5, L. 33: Reinecker et al., 2008 is cited in the text but the reference list only includes Reinecker, et al., 2015.

>>> Revised and corrected for Reinecker et al., 2011.

P. 5, L. 36: Should "weighed" be "weighted"?

>>> Changed to "weighted". Thank you.

P. 5. L. 45: please define TCCON.

>>> Defined as 'The Total Carbon Column Observing Network' in the revision

P. 6, L. 4: recommend adding citation for Jucks, et al., in references list

>>> Added

P. 6, L. 12: suggest changing west side of continents to west side of the United States for clearer context.

>>> Marine stratus clouds also exist at the west side of other continents like South America. So it is a general statement.

P. 6, L. 22: the standard deviations of the two high-altitude and one low-altitude data points appear to be larger than 2-3 ppm as stated, perhaps 2-5 ppm. Can the authors verify this stated range?

>>> Revised to 2-4 ppm

P. 6, L. 23: please define AVOCET

>>> Defined as 'the Atmospheric Vertical Observation of CO2 in the Earth's Troposphere'

P. 6, L. 23: should be "are" after signals

>>> Added

P. 6, L. 40: as stated for line 22 above, the standard deviations of the one low-altitude data point appears to be larger than 3-6 ppm as stated, perhaps 3-10 ppm. Can the authors verify this stated range?

>>> 3-6 ppm except for measurements at the lowest altitude, for the reasons explained earlier.

P. 7, L. 1: need "with" after "results"

>>> Added. Thank you.

P. 8, L. 30: I recommend adding some quantitative results to this section such as accuracy and precision values relative to correlated in situ measurements. The conclusion section is mostly a restatement of the introduction without quantitative results.

>>>Great recommendation. We added the following details in the 3rd paragraph of the section. "Meanwhile, when compared to in situ data with sufficient samples (> 30), the XCO2 retrievals to the puffy cumulus cloud tops near the WBI tall tower showed low bias (~0.2 ppm) and standard deviation of 1.9 ppm. In this case, the standard deviation of XCO2 retrievals to the cumulus cloud tops were increased by 20%, compared to those to the ground of 1.6 ppm, which was caused by the larger cloud top roughness as well as the lower cloud reflectivity at the measurement wavelengths.

P. 8, L. 37: that statement about which cases are excluded due to proximity to cloud tops and aircraft tilt would be useful at the beginning of the manuscript, perhaps in section 2.0.

>>> Agree. We added that in the Section 3.1.

P. 11, L. 23: Ramanathan, et al., 2013 does not appear to be cited in the manuscript text.

>>> It should be cited in Section 2 about measurement approach.

Pgs. 16-17: Figures 3 and 4: are there nadir camera images from the flights that could be added to these figures to provide a visual description of the cloud fields? The pictures on figures 6 and 8 are very helpful for describing the cloud field and should be included with figures 3 and 4 if possible.

>>> Yes, we did have a nadir-viewing camera for all these flights. Clouds for Fig. 4 are actually shown in the left panel of Fig. 6 and clouds for Fig. 3 are similar to the clouds shown in the left panel of Fig. 8.

P. 20, Figure 7: Is the range value in meters here? Suggest adding the unit. What is the definition of the red dotted line (not defined in the legend)? I suggest using thicker lines in the plot for increased readability, and adding the word "dense" before "cloud top" in the caption.

>>> Yes, the range units are in meters. We define all these lines in the figure caption and rewrite it as following,

Figure 7. The lidar transmitted pulse shape (left panel) and the recorded echo pulse shape returned from a dense cloud (right panel). The blue solid lines are for pulse #15 that is near CO2 absorption line center and the red solid lines are for pulse #30 in the line wing. Horizontal black lines are signal baselines. Vertical dashed black lines indicate signal integration windows and the middle dashed red lines are the integration center position in defining the centroid cloud height. The unit of range is in meter.

P. 21, Figure 8: AVOCET should be all caps in the legend. In line 2 of the caption, do the authors mean "altitude intervals" instead of "attitudes"?

>>> Yes, corrected

P. 25, Figure 12: suggest adding "(shown as red dashed line)" at the end of the caption to define the red dashed altitude line in the plot.

>>> Added. Thank you.